PERSPECTIVE

# Making cough count in tuberculosis care

Alexandra J. Zimmer [1,2], César Ugarte-Gil[3,4], Rahul Pathri[5], Puneet Dewan[6],
Devan Jaganath[7,8], Adithya Cattamanchi [7,8], Madhukar Pai[1,2] &
Simon Grandjean Lapierre [2,9,10]✉

**Abstract**

Cough assessment is central to the clinical management of respiratory diseases, including tuberculosis (TB), but strategies to objectively and unobtrusively measure cough are lacking. Acoustic epidemiology is an emerging field that uses technology to detect cough sounds and analyze cough patterns to improve health outcomes among people with respiratory conditions linked to cough. This field is increasingly exploring the potential of artificial intelligence (AI) for more advanced applications, such as analyzing cough sounds as a biomarker for disease screening. While much of the data are preliminary, objective cough assessment could potentially transform disease control programs, including TB, and support individual patient management. Here, we present an overview of recent advances in this field and describe how cough assessment, if validated, could support public health programs at various stages of the TB care cascade.

P rior to the COVID-19 pandemic, tuberculosis (TB) was the leading infectious cause of mortality, resulting in approximately 10.0 million new infections and 1.4 million deaths worldwide in 2019[1]. The COVID-19 pandemic and lockdowns have had a devastating impact on TB programs globally, as resources and tools used to diagnose and manage TB were diverted to COVID-19[2]. To restore progress and mitigate the impact of COVID-19 on TB management, it is essential to leverage new technologies and innovations to improve TB prevention and care.

TB is an infectious disease caused by the inhalation of droplets containing the bacteria *Mycobacterium tuberculosis*[3]. TB varies in presentation, ranging from asymptomatic, non-transmissible TB infection (also known as latent TB infection) to symptomatic, contagious active TB disease[4]. Between these two extremes are subclinical forms of TB, where people are considered asymptomatic but may transmit TB to others[4].

While active TB disease most commonly affects the lungs (pulmonary TB), approximately 15–20% of active TB occurs in other parts of the body, including lymph node TB, abdominal TB, TB meningitis, ocular TB, and neurological TB, to name a few[5]. The occurrence of TB in the body other than the lung is known as extrapulmonary TB (EPTB)[3]. Active pulmonary TB is most commonly diagnosed by microbiological testing on mucus from the lung (sputum) samples. Sputum culture is the gold standard for TB testing. However, it is expensive, slow, and requires access to centralized biosafety laboratories[6]. Sputum smear microscopy is often used in

[1]Department of Epidemiology, Biostatistics and Occupational Health, McGill University, Montreal, Canada. [2]McGill International TB Centre, Montreal, Canada. [3]School of Medicine, Universidad Peruana Cayetano Heredia, Lima, Peru. [4]Instituto de Medicina Tropical Alexander von Humboldt, Universidad Peruana Cayetano Heredia, Lima, Peru. [5]Docturnal Pvt. Ltd., R&D, Hyderabad, India. [6]Bill & Melinda Gates Foundation, Seattle, WA, USA. [7]Department of Medicine, Division of Pulmonary & Critical Care Medicine, University of California, San Francisco, 1001 Potrero Avenue, San Francisco, CA 94110, USA. [8]Center for Tuberculosis, University of California, San Francisco, 1001 Potrero Avenue, San Francisco, CA 94110, USA. [9]Immunopathology Axis, Centre de Recherche du Centre Hospitalier de l'Université de Montréal, 900 Rue Saint-Denis, Montréal, QC, Canada. [10]Department of Microbiology, Infectious Diseases and Immunology, Université de Montréal, 2900 Boulevard Edouard-Montpetit, Montréal, QC, Canada. ✉email: simon.grandjean.lapierre@umontreal.ca

primary care facilities in lower-resource settings as a cheaper alternative, but has low sensitivity and is not able to detect drug-resistance[7]. In recent years, more advanced molecular platforms (e.g., GeneXpert PCR machines) have been scaled up as smear-replacement tools that offer greater sensitivity and quicker turnaround times for TB diagnosis[8,9]. Culture, smear microscopy, and GeneXpert are commonly used as reference standards when evaluating the performance and accuracy of newer diagnostics. While active TB is curable, the long regimens (6 months for drug-susceptible TB) and adverse events caused by the antibiotics used, complicate treatment and increase the risk of drug-resistance emerging[10,11].

As coughing is a common TB symptom, it can be used to diagnose TB and assess effectiveness of treatment. This Perspective discusses advances in acoustic epidemiology and AI-based methods to assess cough and how these can be used during TB diagnosis and treatment.

## Using cough as an objective biomarker for TB control and care

Cough is a complex physiological phenomenon as it is both a symptom of, and a defense mechanism against, respiratory diseases. Cough is a hallmark symptom of pulmonary TB and is clinically assessed throughout the cascade of TB care, for example, as a triage tool to trigger TB testing or to monitor response to therapy. Cough patterns vary depending on the amount of *M. tuberculosis* in the lungs, and cough tends to regress with successful TB therapy[12–15].

While many TB screening programs use cough duration and symptoms to determine when TB testing is required, this symptom screening approach lacks sensitivity. In low-resource settings, peripheral health centers, and communities, triage tools such as chest X-rays are not available, thus symptom-based screening remains the only available strategy to identify people with TB. The World Health Organization (WHO) recommends testing people reporting symptoms compatible with TB, including prolonged cough (usually interpreted as a cough that lasts two weeks or longer)[16]. According to the 2021 WHO TB screening guidelines, the sensitivity of prolonged cough alone is 42% among HIV-negative individuals, well below the WHO community-based triage test target product profile (TPP) of ≥90% sensitivity[16,17].

It is difficult for people to describe their cough symptoms, and it is as challenging for clinicians to identify the cause. Individuals tend to have poor recall of the duration of their symptoms, and symptom severity is subjective[18,19]. Given our current inability to objectively detect and monitor cough sounds, patients and providers systematically reduce this data-rich symptom into subjective and dichotomous information (e.g., cough versus no cough, chronic versus acute, getting better versus getting worse), precluding rigorous understanding of cough data, and preventing the use of cough to its full clinical potential. By making cough an objective and measurable component of TB care, either by helping individuals recognize abnormal cough patterns, or by harnessing artificial intelligence (AI) technology (using computer systems to recognize and interpret the implications of a cough sound)[20] to differentiate types of coughs, we can potentially improve patient management and clinical outcomes at different stages during the cascade of TB care.

## Advances in acoustics for objective cough monitoring

Questionnaire-based tools and scales have been used to collect and evaluate the severity of coughs of varying etiology in an attempt to transform subjective cough reporting into objective data. Such tools include the visual analog scale (VAS), cough symptom score (CSS), and cough diaries[21]. Both the VAS and CSS attempt to quantify the severity of cough based on a patient's perception of their cough. Cough diaries can take various forms, but all depend on patients tracking the frequency and severity of their coughs over time. Other questionnaires expand their assessment of cough to incorporate questions on health-related quality of life[21]. For example, the Leicester Cough Questionnaire (LCQ) is a validated self-completed questionnaire that measures the quality of life of individuals with a chronic cough, and has previously been used to evaluate cohorts of people with TB undergoing anti-TB therapy[22–24]. While such tools are easy to use and implement in clinical settings, they remain subject to bias related to self-perception of health and attention to symptoms, ultimately limiting their clinical application.

Objectivity in cough analysis is improved when using recording devices and computer-assisted acoustic interpretation algorithms. As early as the 1960s, Loudon and Spohn used tape recorders to record and count the coughs of people with TB at night[25]. Other forms of early ambulatory cough meters involved the integration of audio recording devices and electromyogram (EMG) electrodes[26], which simultaneously recorded cough sounds and chest muscle contractions when the person coughed. In 2006, Paul et al. developed and evaluated a self-contained cough monitor composed of an accelerometer (for measuring cough-related vibrations) that stored data on a CompactFlash memory card[27]. This device was attached to the patient's neck in the suprasternal notch (jugular notch) and demonstrated good agreement with coughing seen on video footage. Over the years, more advanced 24 h recording devices have been developed. These devices typically have a microphone (e.g. free-field microphone necklace or one that attaches to the patient's lapel), which sends the cough sounds to a digital sound recorder, usually attached at the hip of the patient[28]. Such recording devices include the Leicester Cough Monitor (LCM), the Cayetano Cough Monitor (CayeCoM), and the VitaloJak[29–31].

Cough counts and patterns were the first objective markers used to analyze cough severity and variation over time. The LCM, CayeCoM and VitaloJak have all been validated for the measurement of cough frequency[29–31]. The LCM and VitaloJak are currently the most widely used cough monitoring tools, with reported cough detection sensitivities of 91% and >99%, respectively[28]. The LCM uses a largely automated algorithm for detecting cough sounds, requiring operator input for calibrating the device (approximately 5 min for every 24 h of recording)[28]. The LCM and the CayeCoM have been used to investigate cough among people with pulmonary TB. Turner et al. used the LCM as part of a cross-sectional survey of cough frequency among people with TB and their contacts[32]. Williams et al. used the LCM to correlate exhaled *M. tuberculosis* with cough frequency[15]. The CayeCoM has been used in various studies to measure cough frequency among cohorts of people with pulmonary TB undergoing treatment[12–14,33,34]. A summary of studies that use various tools for objective cough monitoring in the context of TB care can be found in Supplementary Table 1.

While ambulatory recording devices have enabled continuous cough recording, many of the devices used to date are bulky and obtrusive. Cough is an obvious and stigmatizing symptom, especially among people with TB, and the COVID-19 pandemic has dramatically heightened this stigmatization[35]. In order to efficiently monitor people with cough, recording strategies must be inconspicuous to avoid adding to the stigmatization of respiratory conditions. Smartphones with cough detection and recording applications provide a more discreet approach to monitoring TB coughs. Several cough recording applications have already been developed, including Hyfe Research, AI4COVID-19, and ResAppDx[36–38].

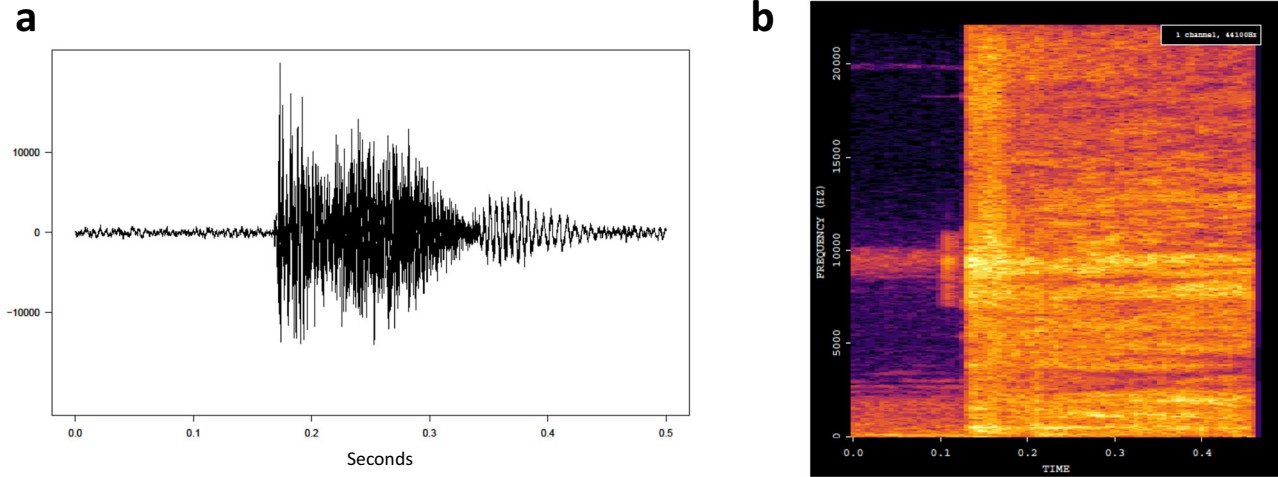

**Fig. 1 Digital cough spectrograms for artificial intelligence algorithm analysis. a** Waveform image of a pulmonary TB cough. **b** Spectrogram conversion of the waveform cough. On the spectrogram, acoustic information is represented as frequency (*y*-axis) and amplitude (color) over time (*x*-axis).

## Developments in artificial intelligence allow for rigorous assessment of cough

Advances in machine learning, a subset of AI that enables machines to apply algorithms on available data to automatically "learn" and make autonomous decisions[20], has given rise to a variety of algorithms for cough monitoring that can be deployed on digital recording devices, including smartphones (see Supplementary Table 1 for examples of the types of algorithms used for cough detection and cough classification). This new technology allows the analysis of both the frequency and the nature of cough sounds. For example, some algorithms first transform sound recordings into spectrograms—a visual representation of the frequency, amplitude, and time characteristics of sounds—before running an algorithm on the spectrogram to visually analyze the cough's features (Fig. 1).

These algorithms are being trained to identify human coughs from ambient sounds (cough detection), as well as to differentiate coughs from patients with distinct clinical conditions or at different stages of disease (cough classification), though the latter use case is yet to be validated[39–43]. Several preliminary cough classification algorithms have been developed for COVID-19 and TB. A classification algorithm was reported to detect COVID-19 infections among people with a cough with 98% sensitivity and 94% specificity, based on a sample of 5320 individuals (half of whom were COVID-19 positive) and against a reference standard of an "official test" (laboratory method accepted as a diagnosis for COVID-19), doctor assessment, or personal assessment[42]. Another group reported that COVID-19 could be diagnosed using cough with 89% sensitivity and 97% specificity[37]. For TB, TimBre is a screening application that leverages machine learning to detect TB coughs with a sensitivity of 80% and specificity of 92% against a composite reference standard of sputum smear microscopy, GeneXpert, and chest X-ray, from a sample of 5 bacteriologically-positive and 469 bacteriologically-negative individuals[44]. Another study developed a cough-based screening system that could discriminate cough sounds produced by 16 individuals with TB from those produced by 35 individuals with other lung diseases with 93% sensitivity and 95% specificity against a bacteriological (laboratory method not specified) reference standard, achieving the WHO's TPP requirements of 90% sensitivity and 70% specificity for a community-based TB triage test[3,45]. Botha et al. also developed an AI algorithm for TB cough classification from a sample of 17 people with TB and 21 healthy individuals, achieving an accuracy of 78% and a sensitivity of 95%, at a specificity of 72%

against a sputum culture reference standard[46]. These early studies demonstrate that digital cough monitoring, including detection and classification of cough events, could potentially be used to assist TB screening (see Supplementary Table 1). However, further development and evaluation is critical to move the field forward.

The accuracy of these AI algorithms is contingent on the characteristics of the training dataset. To date, external validation of various AI algorithms has been limited, or has not yet been performed, and the sample sizes used to evaluate these algorithms have been relatively small[47]. Additionally, early diagnostic studies of novel tests, including AI algorithms, tend to overestimate the diagnostic accuracy, mainly because of the preferential exclusion of more complicated cases[48]. Until sufficient replication studies have been completed using large, and diverse cough datasets, representative of different populations, the clinical application of these AI algorithms will remain limited.

## Using digital cough monitoring to change TB care

Digital cough monitoring has the potential to address multiple gaps in the TB cascade of care (Fig. 2)[49]. In this section, as an example of the breadth of the potential value of cough data, we outline hypothetical ways in which AI-based cough tools could be used.

**Supporting TB program planning.** Finding people with TB, or who have symptoms of TB, requires health systems and TB programs to strategically deploy limited resources. In a syndromic surveillance approach (i.e. detection and aggregation of individual and population health indicators, such as symptoms, prior to establishing a definitive diagnosis) both individuals at risk of developing TB, or people who previously had TB, could passively and prospectively monitor their cough. Temporal and geospatial aggregations of cough events could in turn be used to better target case-finding activities and identify high-risk settings. Spatio-temporal changes in cough frequency at the population-level can be used as a proxy for the incidence of COVID-19, TB or other respiratory diseases[36]. Whether specifically dedicating public health resources to investigate such cough clusters would accelerate the identification of additional prevalent cases and improve disease case notifications needs to be investigated. Restricting this cough surveillance analytic approach by monitoring people previously diagnosed active pulmonary TB could identify cough hotspots where the risk of TB transmission has been, and may still be, even higher.

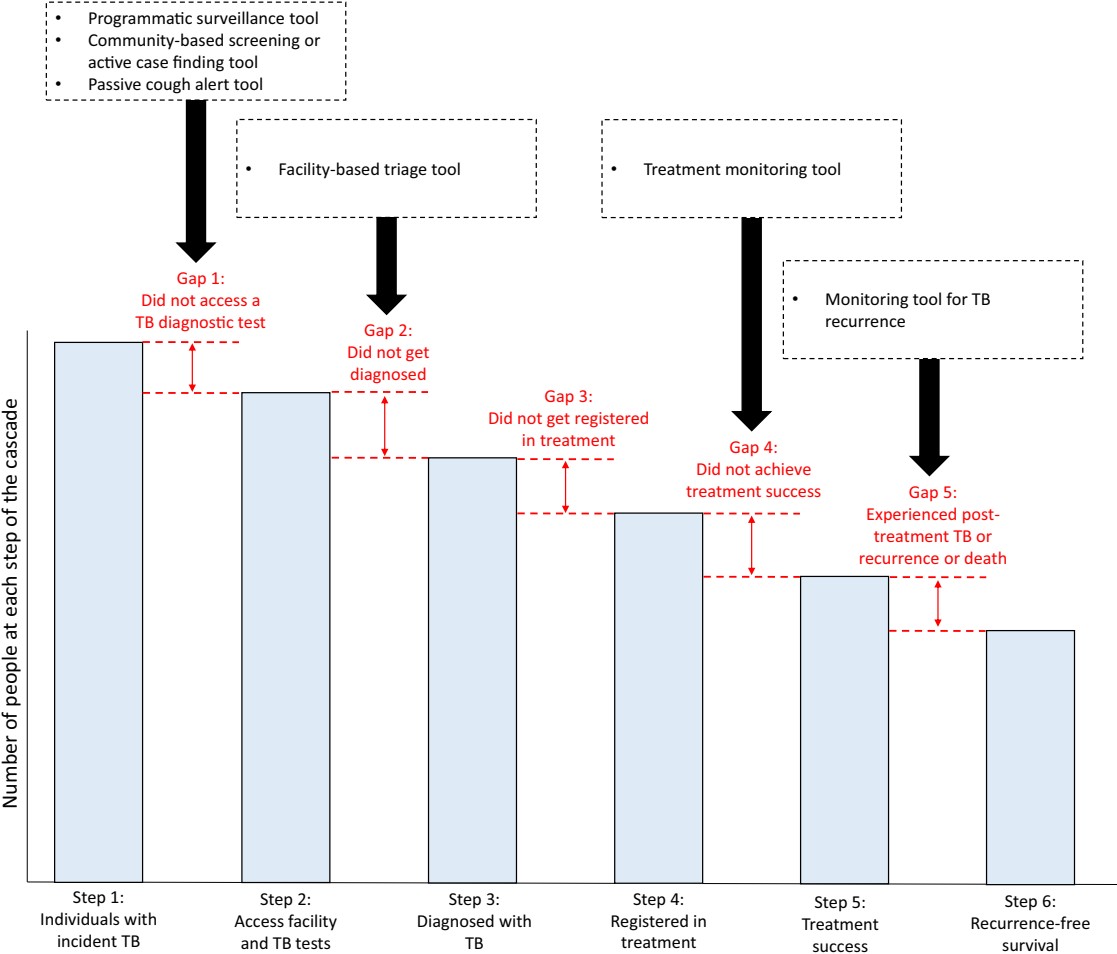

**Fig. 2 Potential use cases for digital cough monitoring in the tuberculosis cascade of care.** Each step in the TB care cascade is represented as a bar. The gaps in the cascade are in red between each step. Boxes pointing at the gaps represent possible digital cough-based solutions to address various gaps. The height of the bar graphs and the length of the gaps are not scaled to represent true values. They are intended to help illustrate the different steps of the care cascade and points at which people with TB may fail to benefit from care. (Cascade of care adapted from Fig. 1 of Subbaraman et al.)[49].

**Improving community-based monitoring and active case finding.** Very preliminary data suggests that cough classification algorithms could be developed that meet WHO TPPs for a community-based TB screening test[44,45]. Further validation is needed using cohorts of large sample size and diverse populations before any definite conclusions can be made regarding their sensitivities and specificities. AI-based cough screening could complement other available community-based screening approaches, such as chest X-rays, increasing the number of people with presumed TB appropriately referred to facilities for confirmatory testing in a timely manner. Indeed, using cough to predict chest X-ray abnormalities could trigger radiology testing for which multiple automated interpretation algorithms have now been thoroughly validated[50]. If deployed on mobile devices, AI-based cough screening could allow for low-cost remote active case finding and self-screening, with subsequent referral to a health facility for confirmatory TB testing and linkage to care. The vignette in Fig. 3 illustrates how a cough monitoring tool may help refer people with a cough to a physician.

For individuals at higher risk of developing active pulmonary TB, such as household contacts, cough detection and longitudinal monitoring could objectively document an increase in TB-compatible symptoms, prompting early care-seeking and limiting transmission. This approach could also help address subclinical pulmonary TB[51]. Individuals who have mild symptoms, but do not recognize them as being significant, are also considered subclinical[51]. In such cases, digital cough monitoring could be used to identify the presence or significance of cough that would otherwise have gone unrecognized or unreported. However, digital cough monitoring would not extend to truly asymptomatic individuals with subclinical TB, limiting its application as an active case finding tool in this sub-group. A study of 24 people with TB found that cough frequency may not be associated with *M. tuberculosis* output collected on face masks[15]. That is, some participants who did not cough very often still expelled a lot of *M. tuberculosis* (and vice versa). While further investigations are needed, this raises potential limitations of relying on cough monitoring for evaluating active case-finding and reducing TB transmission.

**Enhancing the performance of diagnostic algorithms.** Even when people with presumed TB reach the health facility, it is not guaranteed that they will access proper confirmatory testing. One reason for this is a lack of awareness and training among healthcare workers to recognize key TB symptoms. This problem has been demonstrated by studies involving standardized patients (SPs), healthy persons trained to visit health facilities with fake TB symptoms, without the healthcare providers being aware that these symptoms are not real[52]. A systematic review on SPs in India found that only half of healthcare providers knew that prolonged cough (>2 weeks) may be associated with TB[52].

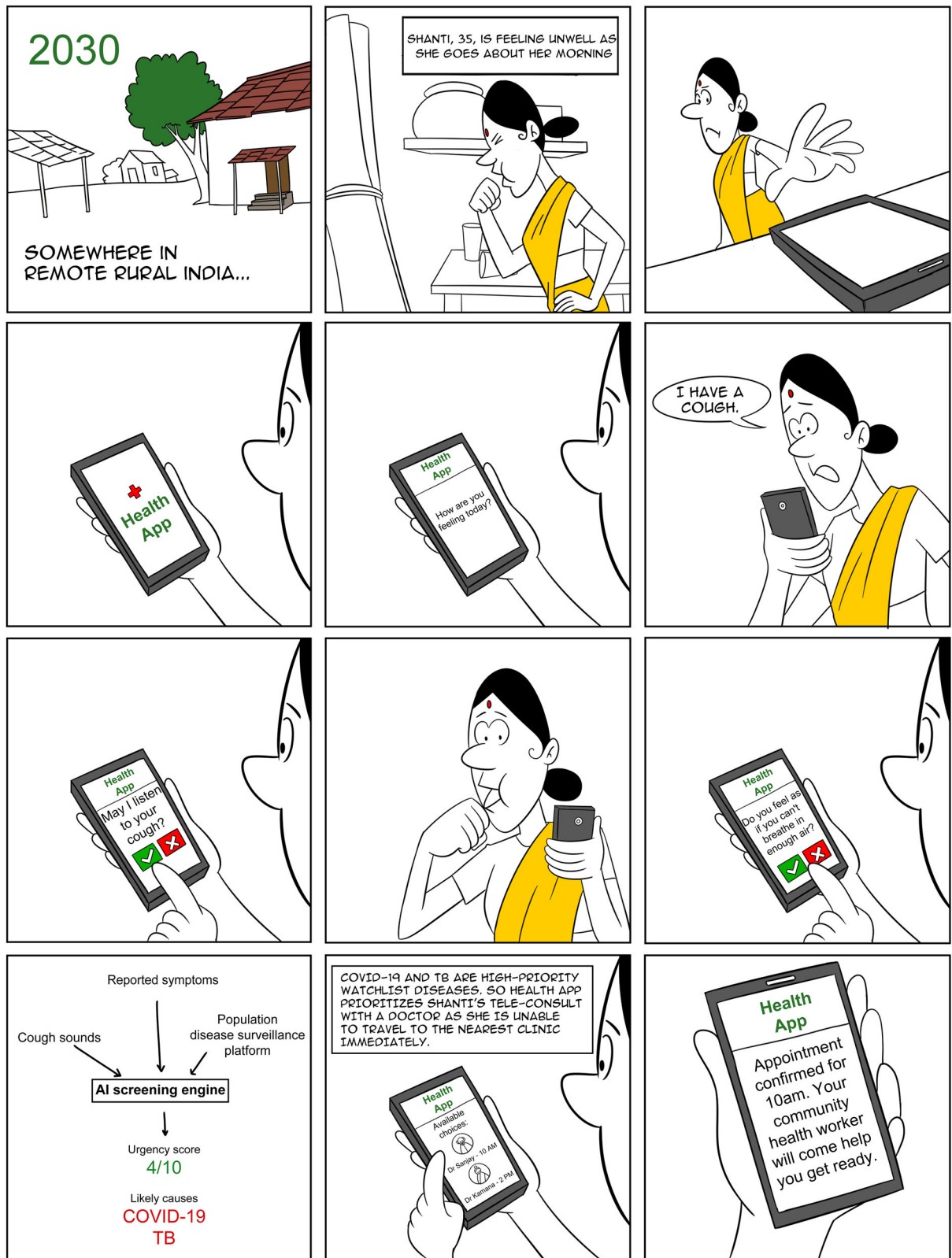

**Fig. 3 Example use of smartphone-based cough screening application for community-based monitoring.** In this vignette, a female is experiencing symptoms of disease, including cough. Using a phone with the example Health App (not a real app), she is prompted to cough and report any other symptoms she is experiencing. The AI algorithm in the Heath App uses the information to provide likely causes of disease (in this case, COVID-19 or TB) and refers her to consult a physician for confirmatory testing. (Vignette originally created for *The Lancet Citizens' Commission on Reimagining India's Health System*, by Raghu Dharmaraju, Vijay Chandru, Umakant Soni, and Shubraneel Ghosh, ARTPARK (AI & Robotics Technology Park) at Indian Institute of Science. "*A vignette from 2030 in rural India: How might technology enable citizen-centered health journeys?*" https://www.artpark.in/reimagine-health/2030_rural_india).

Another study in India found that SPs presenting with TB symptoms were severely under-tested[53].

Similar to community-based screening and active case-finding, health providers may potentially use AI-based cough classification applications to help triage people with presumed TB, complementing less sensitive symptom-based triage methods and increasing the proportion of individuals with presumed TB who undergo confirmatory testing. Because symptom screening is also non-specific, cough classification tools may also help reduce the proportion of people without TB who unnecessarily undergo TB testing.

**Monitoring the effect of treatment**. Smartphones are globally available and can act as recording devices. They are already used for TB treatment-adherence monitoring with video Directly Observed Therapy (vDOT), which allows people with TB to send videos of themselves ingesting anti-TB treatment to their health provider, instead of having to travel to the clinic to take their anti-TB treatment in front of a health provider, as required under traditional DOT methods[54]. Given that cough symptoms regress with successful treatment, cough detection applications could be used as a low-cost, person-centric approach for clinicians to remotely monitor people with TB's clinical response to treatment, or even for people to self-monitor their cough as treatment progresses[13]. Objectively-documented unfavorable cough evolution patterns could prompt patients and providers to investigate whether the treatment regimen being used is effective, allowing for early recognition of drug resistance or poor adherence.

**Achieving relapse-free cures and minimizing long-term lung damage**. A significant proportion of people who are successfully cured of TB are at risk of TB recurrence within the first year following completion of anti-TB treatment[55]. The prospective cough monitoring used during treatment could be continued during this high risk period to identify early signs of TB recurrence. Even if people do not experience TB recurrence or relapse, they are at increased risk of experiencing post-TB lung damage, an aspect of TB care that is often overlooked in TB management pathways[56]. Thus, cough monitoring, if validated, could also be useful as a starting point in identifying individuals with post-TB lung disease and related lung function decline.

**Supporting drug development and TB research**. AI-based cough detection technology could also play a role in TB research and development. Digital cough monitoring could be used as a secondary endpoint in clinical drug development trials. Drug development trials have so far relied on evaluating whether sputum culture test results change from positive to negative during the first 8 weeks of therapy as a proxy for anti-TB treatment efficacy[57]. Such culture methods are resource- and time-intensive, and do not allow the monitoring of intermediate outcomes, including patient symptoms. In addition, regulatory agencies may request data on patient-reported improvement in cough, though again this is subjective and can have variable accuracy[58,59]. Similar to symptom-based screening, self-assessment of cough in the context of experimental therapy-efficacy measurement is unlikely to be fully accurate. Objective monitoring of cough may allow for more nuanced monitoring of intermediate endpoints by acting as a complement to conventional culture-based endpoints and patient-reported outcomes.

### Furthering the clinical use of digital cough monitoring

The recent progress in acoustics and cough analysis, combined with the urgent need to improve respiratory disease detection and tracking methods in the context of COVID-19, have accelerated applications of acoustic epidemiology in clinical research[37,42,60]. This emerging field depends on increasingly less obtrusive ways to collect cough data as well as more sophisticated analytics that go beyond cough detection to infer clinical etiology based on cough patterns and spectral characteristics.

The development, validation, and roll-out of digital cough monitoring tools for TB will require global coordinated data collection, curation, and analysis effort. Training and validation cough datasets need to be collected from people in the intended use population and settings. They must include large numbers of people with different demographic characteristics (e.g. age, sex, ethnicity) as well as different forms of pulmonary TB in clinical settings with variable background epidemiology of respiratory diseases. This 'big data' approach is mandatory for the development and refinement of AI algorithms to achieve high external validity. Since cough is not specific to TB, such datasets should not be limited to the development of AI algorithms for TB but should also be used to develop and refine cough algorithms for other respiratory diseases and conditions that are linked to cough. To accelerate this endeavour, we must avoid the multiplication of isolated algorithm development efforts that use data from homogeneous patient populations[47].

Collective efforts to aggregate and annotate cough data may accelerate research and tool development. For example, Global Health Labs, the Bill and Melinda Gates Foundation, and the Patrick J. McGovern Foundation are currently supporting efforts to collect cough data and are investing in infrastructure to build an extensive database of cough sounds. Researchers interested in cough and acoustic epidemiology—in the context of TB or any other respiratory disease or condition linked to cough—can contribute to this growing anonymized database and use the existing data to develop and refine AI. While this effort is an important step towards integrating cough into TB care, there is still a need for a broader recognition of the potential advantages of integrated AI-based cough tools into TB care. As more AI-based cough detection tools and applications become available, increased effort should be made to routinely collect cough data within TB programs, prevalence surveys, and clinical studies in order to contribute to the growing field of acoustic epidemiology. Such efforts will help characterize the natural evolution of TB cough, objectively describe the impact of specific interventions on TB symptoms, and iteratively improve operational and performance characteristics of cough-based TB solutions. Like other biomarkers, collected cough data must be anonymized, annotated with clinical metadata, and shared in open-source repositories. TB cough data must also be made available in the same way that digital chest X-ray libraries are available for the validation of electronic interpretation algorithms, or that TB genomic sequences are available to support novel drug development and validation of drug resistance assays[61,62]. Through such collective efforts, we can accelerate algorithm development and the roll-out of cough-based clinical tools. This data sharing approach should also improve partnerships between academia and industry by allowing faster hypothesis-testing as well as rapid product design and translation into user-friendly tools that can be deployed at scale in TB care. In conclusion, AI and acoustic epidemiology have the potential to revolutionize the fight against TB.

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

## Acknowledgements

We would like to thank all people with TB, providers and researchers who have and will contribute acoustic data to the tuberculosis scientific community to make cough count in tuberculosis care. We would also like to thank Raghu Dharmaraju, Vijay Chandru, Umakant Soni, and Shubraneel Ghosh from ARTPARK (AI & Robotics Technology Park) at the Indian Institute of Science for allowing us to use the vignette they developed that illustrates the use of smartphone-based cough detection for community-based screening.

## Author contributions

Conceptualization (A.J.Z., S.G.L.) literature article review (A.J.Z., D.J., S.G.L.), cascade of TB care design and assessment of clinical applications (A.J.Z., C.U.G., R.P., P.D., D.J., A.C., M.P., S.G.L.), machine learning perspective and tuberculosis digital cough analysis (R.P.), collaborative cough consortium design and funding acquisition (P.D., A.C., S.G.L.), writing original draft (A.J.Z., S.G.L.), writing review and editing (C.U.G., R.P., P.D., D.J., A.C., M.P.), visualization (A.J.Z.).

## Competing interests

The authors declare no competing interests.
