## [Peer Review File · Communications Medicine]

Reviewers' comments:

Reviewer #1 (Remarks to the Author):

This is an interesting, timely, and important contribution. It is interesting, as it suggests a number of well-reflected use cases for AI-based cough recognition and monitoring for the application field of TB. It is timely, as with the COVID-19 outbreak, significant efforts came from the Computer Audition community to improve on coughing recognition performance. At the same time, deep learning advances helped from a technical end to render such systems mostly ready for real-world usage. And, it is important, as it links these results and developments to the medical readership community.

That said, the review part is held fairly short, and it is doubtful of how much use the sheer mentioning of CNNs is to the reader. In fact, audio signals can and are also presented in 1D in some approaches (as opposed to the mention that they would always be transformed into 2D by some time-frequency transformation). In addition, many other approaches exist, reaching from traditional feature representations developed over decades by experts to other deep approaches such as by recurrent sequence-to-sequence autoencoders for representation learning.

Further, while the authors give some short word of warning on how to interpret the results named in coughing recognition papers, one might strengthen this part. In real-world application, such rates can and usually do fall quite behind.

What would likely be a great addition to the present manuscript is some table summarizing different approaches and results as for AI-based coughing recognition. In fact, even the nicely suggested use-cases might be summarized in another table.

Recognition rates might be presented in a unified way as to precision, i.e., not once XX%, once XX.XX%.

Otherwise, a very well written manuscript that can easily inspire beyond the scope of application in TB.

Reviewer #2 (Remarks to the Author):

This is a very interesting and timely perspective that I suggest to accept for publication.

I do have a few suggestions and points for clarification.

I understand this is a Perspective and not a Review paper. In the format of a Perspective ideas can be discussed from a personal viewpoint and they may be opinionated. However, I do think the paper would be stronger if it also has a section providing a more critical view on existing evidence and thoughts on practical implementation.

line 53-54: 'Many coughers do not seek care' seems to be the explanation for the poor sensitivity of cough as a screening tool mentioned in line 53. (this because the latter part of the sentence 'most people who cough do not have TB' relates to the specificity). Health seeking behavior is indeed often a problem when relying on passive case finding, but I would not link it to the sensitivity of cough as a

screening tool.

line 60: true the sensitivity of using a cough of 2 weeks or more as screening tool is only 42% (with spec of 94%), however using any symptom as TB has a sens of 71% (at the cost of a much lower spec of 64%). It is not entirely clear where 'this approach' in line 60 refers to. To me it refers to 'reporting symptoms compatible to TB, including prolonged cough' and in that case the 71% is the sens to mention.

line 61-62. I do not fully grasp this statement. What do you mean to say and do you have any reference for this statement?

line 66: do you have a reference for this statement 'Individuals tend ... their symptoms'?

line 129-131: when reading these lines, I felt more details are needed on what evidence is available that shows algorithms can differentiate cough with distinct clinical conditions. And whether there have there been studies exploring whether TB can be one of the diseases to be distinguished. Later I realized these details are presented in lines 139-151. Why not connect these parts and remove the technical details presented in rows 131-137 which in general do not add a lot?

lines 143-145: could you provide a few more details on this tool and its clinical validation? in what population was this study done? Why was this data not published?

lines 146-149: these results are definitely promising. Good to mention these were based on 16 TB patients and 35 non-TB patients.

lines 160-162: sentence seems incomplete and/or there is a typo (imroe)

line 164: o=of?

line 170-171: I do not understand how cough events from confirmed TB patients could be mapped (retrospectively?). I initially also had difficulties understanding how cough monitoring of general populations could be used to identify high-risk settings. Do you suggest to use cough surveillance as a proxy for measuring TB prevalence (and other respiratory diseases) and thus with large enough sample size (depending on the underlying TB prevalence)? That sounds like a great idea.

row 178: very preliminary data I would say..

row 182-183: could this also be used as a self-screening tool? with subsequent referral to a health facility for TB testing?

row 206: could this be done at the home of a patient by the patient themselves?

row 255: this database that is being developed, it is focused on TB cough or does it have a much wider scope? In general the last section (from row 242 onwards) focusses very much on collecting cough data for TB tool development, which is logical as this is the focus of the perspective. Still, people/patients want to know what the cause is of their cough and when it is not TB they want to know what it is instead. I wonder whether this is something that can be addressed, to take a less vertical view/approach.

line 353: reference 21 is incomplete; 27 as well

Reviewer #3 (Remarks to the Author):

A perspective on the potential use of recording of cough analysed by Artificial Intelligence to the detection, specific diagnosis, and monitoring of treatment of tuberculosis is presented. The conclusion is that such application could revolutionise tuberculosis care.

The following comments are for the authors' consideration

1. The reviewer agrees this is an interesting idea that has been advanced by the COVID-19 epidemic. However, the cited evidence for the use of cough in those circumstances in this article is slim. Perhaps a table of key studies would help more persuasively make the case this technique should be more widely reapplied to tuberculosis.
2. The authors do not really set out much of a blueprint for research other than to say data collection should be in some way standardized and shared. Yet they mention a few specific applications in detection, diagnosis and treatment monitoring that could be compared with standard of care using rigorous trial designs to determine whether there was incremental advantage. The tuberculosis control world is, in general, not led by early adopters and simply saying something should be massively implemented and analysed because it is possible and cheap may not cut the cheese as evidence for some who determine policy.
3. Line 111 sensitivity for what?
4. Line 143 – 145 It may not be considered adequate to reference a statement that the TB TimBre has diagnostic utility by mentioning this is based on an unpublished third-party clinical validation. The reader is left wondering what this means and has no means to evaluate this evidence nor even to glean what the perspective authors think of it.
5. Several spelling mistakes lines (lines 162, 187)
6. Several references incomplete (9, 23, 27, 29, 37, 39)

Reviewer #4 (Remarks to the Author):

This is an interesting manuscript evaluating the potential role for quantitative cough analysis (using AI) as a measure to be used in TB clinical care, epidemic control and research. The authors make the argument that cough is crudely utilised at present (usually as a subjective question recorded as a binary variable) and that much valuable information may be lost in the process. They present studies that have been done to assess diagnostic performance of cough for TB screening and diagnosis, but also discuss the limitations of this data and the need for external validation studies. They also call for collaborative efforts to combine data sets and harmonise data collection, to allow AI to be used to further development of this nascent field. They discuss where in the TB cascade cough measures may have an impact. Overall, a useful summary of a research area that needs to be pursued further for its potential.

The authors should discuss the limitation that many patients have subclinical TB and are not coughing – around 50% in a recent prevalence study in South Africa. While they do mention that among people with subclinical TB cough measures may help to detect under-reported cough, its

likely that most people in this category do not have a cough. This should be stated as a recognised limitation of cough's utility of detecting undiagnosed TB in the community.

For the studies described in line 139-151, it would be useful to provide more details of these studies in a supplementary table – what was the design of these studies, sample size, reference standard, if it was case-control what were controls, etc. It should also be emphasised in this paragraph that early diagnostic studies of novel tests always tend to overestimate diagnostic performance.

Line 171 – its unclear how mapping “cough events from confirmed TB patients” would help with TB control efforts. Please explain this.

Line 210-211. This assumes that cough is the only mechanism via which the infection is transmitted. There is emerging evidence that other respiratory manoeuvres also transmit TB. This should be noted as a potential limitation.

There are several spelling errors in the manuscript – eg. line 162

Note: line numbers refer to the track changes version.

Reviewer #1 (Remarks to the Author):

This is an interesting, timely, and important contribution. It is interesting, as it suggests a number of well-reflected use cases for AI-based cough recognition and monitoring for the application field of TB. It is timely, as with the COVID-19 outbreak, significant efforts came from the Computer Audition community to improve on coughing recognition performance. At the same time, deep learning advances helped from a technical end to render such systems mostly ready for real-world usage. And, it is important, as it links these results and developments to the medical readership community.

We thank Reviewer #1 for their thoughtful comment and for taking the time to review our paper.

That said, the review part is held fairly short, and it is doubtful of how much use the sheer mentioning of CNNs is to the reader. In fact, audio signals can and are also presented in 1D in some approaches (as opposed to the mention that they would always be transformed into 2D by some time-frequency transformation). In addition, many other approaches exist, reaching from traditional feature representations developed over decades by experts to other deep approaches such as by recurrent sequence-to-sequence autoencoders for representation learning.

In light of the comment raised by Reviewer #2 regarding this section (“remove the technical details presented in rows 131-137 which in general do not add a lot”), we have decided to remove the details concerning CNN and made the discussion of the AI methods broad and short. The focus of this piece is not a review of audio processing methods. The updated text reads as such: “Advances in audio analysis methods and machine learning have given rise to a variety of AI algorithms for cough monitoring that can be deployed on digital recording devices, including smartphones (see Appendix 1 for examples of the types of AI algorithms used for cough detection and cough classification). This new technology allows for the analysis of both the frequency and the nature of cough sounds. For example, some algorithms first transform sound recordings into spectrograms—a visual representation of the frequency, amplitude, and time characteristics of sounds—before running an algorithm on the spectrogram to visually analyze the cough’s features (Figure 1).” (LINE 188-203)

As part of the table in the appendix, we have included a list of some of the different types of AI algorithms used by different papers, both for cough detection and cough classification (TB and COVID-19). This provides a short summary for readers interested in seeing what algorithms have been used before. We directly refer readers to the appendix in the main body of the text: “(see Appendix 1 for examples of the types of AI algorithms used for cough detection and cough classification)” (LINE 191 – 192).

Further, while the authors give some short word of warning on how to interpret the results named in coughing recognition papers, one might strengthen this part. In real-world application, such rates can and usually do fall quite behind.

We agree with the author and have included further warnings regarding the interpretation of novel AI algorithms: “The accuracy of these AI algorithms is contingent on the characteristics of the training dataset. To date, external validation of various AI algorithms has been limited or not been performed and the sample sizes used to evaluate these algorithms have been relatively small. Additionally, early diagnostic studies of novel tests, including AI algorithms, tend to overestimate the diagnostic accuracy,

mainly because of the preferential exclusion of more complicated cases” (LINE 231-236).

What would likely be a great addition to the present manuscript is some table summarizing different approaches and results as for AI-based coughing recognition. In fact, even the nicely suggested use-cases might be summarized in another table.

As mentioned above, we have now included a column in the Table 1 appendix that contains a summary of some of the AI approaches used for cough detection and cough classification. We also directly refer to that table in the main body of our text: “(see Appendix 1 for examples of the types of AI algorithms used for cough detection and cough classification)” (LINE 191 – 192).

We believe that our Figure 2 summarizes the use-cases. We do not think a table for use-cases will add much value.

Recognition rates might be presented in a unified way as to precision, i.e., not once XX%, once XX.XX%.

We have made the suggested change by standardizing to XX% throughout the paper.

Otherwise, a very well written manuscript that can easily inspire beyond the scope of application in TB.

Thank you for your review.

Reviewer #2 (Remarks to the Author):

This is a very interesting and timely perspective that I suggest to accept for publication.

We thank Reviewer #2 for their support and for reviewing our paper.

I do have a few suggestions and points for clarification.

I understand this is a Perspective and not a Review paper. In the format of a Perspective ideas can be discussed from a personal viewpoint and they may be opinionated. However, I do think the paper would be stronger if it also has a section providing a more critical view on existing evidence and thoughts on practical implementation.

We agree that a more critical review of the existing evidence would strengthen our manuscript. We have included a summary table in the Appendix that summarizes the different cough studies discussed in our manuscript. The table summarizes various topics covered in this manuscript: 1) how objective cough assessment has been used in the context of TB surveillance and care; 2) promising AI tools and algorithms for cough detection; and 3) preliminary study results for COVID-19 and TB cough classification AI algorithms.

In addition, we have included more text within the manuscript describing these studies. For objective cough monitoring for TB: “The LCM uses a largely automated algorithm for detecting cough sounds, requiring operator input for calibrating the device (approximately 5 minutes for every 24 hours of recording). The LCM and the CayeCoM have been used to investigate cough among cohorts of pulmonary TB patients. Turner et al. used the LCM as part of a cross-sectional survey on cough frequency among TB patients and their contacts. Williams et al. used the LCM to correlate exhaled TB

bacillary with cough frequency. The CayaCoM, developed by a TB research group in Peru, has been used in various studies to measure cough frequency among cohorts of pulmonary TB patients undergoing treatment. A summary of studies that use various tools for objective cough monitoring in the context of TB care can be found in Appendix 1.” (LINE 162-171)

More detail in the cough classification studies has also been added, including information on the sample sizes used and the reference standards against which these are being compared (this information is also included in the Appendix table): “Another group from Michigan State University reported that COVID-19 could be diagnosed using cough with 89% sensitivity and 97% specificity, based on a sample of 5,320 individuals and against a reference standard of an “official test”, doctor assessment, or personal assessment. Another study developed a cough-based screening system that could discriminate cough sounds produced by 16 TB patients from those produced by 35 patients with other lung ailments with 93% sensitivity and 95% specificity against a “bacteriological” reference standard, achieving the WHO’s TPP requirements of 90% sensitivity and 70% specificity for a community-based TB triage test. Botha et al. also developed an AI algorithm for TB cough classification from a sample of 17 TB patients and 21 healthy controls, achieving an accuracy of 78%, and achieving a sensitivity of 95% at a specificity of 72% against a sputum culture reference standard.” (LINE 211-226)

Regarding the Reviewer’s final point on practical implementation, we believe that the section “Using digital cough monitoring to change TB care”, which outlines the different ways cough tools could be implemented and used in the context of TB care, sufficiently addresses this. We highlight that, given the state of knowledge, much of this is hypothetical and should be further explored.

line 53-54: ‘Many coughers do not seek care’ seems to be the explanation for the poor sensitivity of cough as a screening tool mentioned in line 53. (this because the latter part of the sentence ‘most people who cough do not have TB’ relates to the specificity). Health seeking behavior is indeed often a problem when relying on passive case finding, but I would not link it to the sensitivity of cough as a screening tool.

We agree with the reviewer that this sentence is a bit misleading. We have decided to keep the focus of this paragraph on the lack of sensitivity of cough as a TB screening tool since according to the WHO screening guidelines it meets the target product profile recommendation for specificity (>80%). We have therefore deleted the sentence highlighted by the reviewer and modified the previous sentence: “While TB screening programs still heavily rely on cough and symptoms to trigger TB testing, this symptom screening approach lacks sensitivity and is challenging to deploy in a systematic and sustained way at population level” (LINE 96-99)

line 60: true the sensitivity of using a cough of 2 weeks or more as screening tool is only 42% (with spec of 94%), however using any symptom as TB has a sens of 71% (at the cost of a much lower spec of 64%). It is not entirely clear where ‘this approach’ in line 60 refers to. To me it refers to ‘reporting symptoms compatible to TB, including prolonged cough’ and in that case the 71% is the sens to mention.

We agree with the reviewer that some clinical settings would use any TB symptom for screening, however prolonged cough alone can also be used as a screening tool. This is highlighted in the Annex 1 of the “WHO Operational Handbook on TB: Systematic Screening for TB Disease” where different screening algorithms are shown. Algorithms A.1.1-A.1.4 rely on prolonged cough alone. We have included one of the algorithms as an example below. We therefore do not think we need to refer to the combined symptom approach with 71% sensitivity in order to keep the focus centered on the use of

cough and not broader TB symptoms. We have however replaced “this approach” with “prolonged cough alone” (LINE 106) to make sure it is clear to the readers what type of screening algorithm we are referring to.

Fig. A.1.1 – Screening with cough

line 61-62. I do not fully grasp this statement. What do you mean to say and do you have any reference for this statement?

We have reworked this statement and moved it to the intro sentence of the paragraph: “this symptom screening approach lacks sensitivity and is challenging to deploy in a systematic and sustained way at population level” (LINE 97-99).

line 66: do you have a reference for this statement ‘Individuals tend ... their symptoms’?

We have now included two references (Cho et al. 2019 and Feikin et al. 2010) that discuss the poor recall of symptom duration and the subjectivity of cough severity. We have rephrased this sentence slightly to ensure it is more aligned with the references: “Individuals tend to have poor recall of the duration of their symptoms and symptom severity is subjective” (LINE 114-115)

line 129-131: when reading these lines, I felt more details are needed on what evidence is available that shows algorithms can differentiate cough with distinct clinical conditions. And whether there have there been studies exploring whether TB can be one of the diseases to be distinguished. Later I realized these details are presented in lines 139-151. Why not connect these parts and remove the technical details presented in rows 131-137 which in general do not add a lot?

We have decided to remove the details concerning CNN and made the discussion of the AI methods broad and short. This also allows us to connect with the next section about cough classification for TB, as highlighted by the Reviewer. The updated text reads as such: “Advances in audio analysis methods and machine learning have given rise to a variety of AI algorithms for cough monitoring that can be deployed on digital recording devices, including smartphones (see Appendix 1 for examples of the types of AI

algorithms used for cough detection and cough classification). This new technology allows for the analysis of both the frequency and the nature of cough sounds. For example, some algorithms first transform sound recordings into spectrograms—a visual representation of the frequency, amplitude, and time characteristics of sounds—before running an algorithm on the spectrogram to visually analyze the cough’s features (Figure 1).” (LINE 188-203)

lines 143-145: could you provide a few more details on this tool and its clinical validation? in what population was this study done? Why was this data not published?

The publication for TimBre was recently posted in medrxiv. We have included the reference and updated the information presented to reflect the manuscript. We have decided to include different clinical validation values of the tool TimBre that were published: “In TB, TimBre is a screening application that leverages machine learning to detect TB positive coughs with a sensitivity of 80% and specificity of 92% against a composite reference standard of sputum smear microscopy, GeneXpert (unspecified cartridge), and chest X-ray from a sample of 5 TB-positive and 469 TB-negative individuals” (LINE 214-218)

lines 146-149: these results are definitely promising. Good to mention these were based on 16 TB patients and 35 non-TB patients.

We agree with the reviewer that the sample should be mentioned. We have revised the sentence to include the number of TB and non-TB patients: “Another study developed a cough-based screening system that could discriminate cough sounds produced by 16 TB patients from those produced by 35 patients with other lung ailments” (LINE 219-221)

lines 160-162: sentence seems incomplete and/or there is a typo (imroe)

We have deleted this sentence (LINE 242-245).

line 164: o=of?

We have changed it to “of”.

line 170-171: I do not understand how cough events from confirmed TB patients could be mapped (retrospectively?). I initially also had difficulties understanding how cough monitoring of general populations could be used to identify high-risk settings. Do you suggest to use cough surveillance as a proxy for measuring TB prevalence (and other respiratory diseases) and thus with large enough sample size (depending on the underlying TB prevalence)? That sounds like a great idea.

Mapping would be done prospectively. The idea would be that a population would have a cough tracking application on their smartphone that continuously and passively records their coughs. This is then uploaded to a database and, using phone GPS functionalities, can map out both the geospatial (where) and temporal (when) patterns of coughs. This could be deployed in communities or regions to identify where active case finding activities should be concentrated. We have restructured this paragraph to make it more clear how cough could be used as a surveillance tool: “Finding TB patients requires health systems and TB programs to strategically deploy limited resources. In a syndromic surveillance approach, both TB susceptible individuals or previously confirmed TB patients could passively and prospectively monitor their cough. Temporal and geospatial aggregations of cough events

could in turn be used to better target case finding activities and identify high-risk settings. Temporospatial changes in cough frequency at population-level can be used as a proxy for COVID-19, TB or other respiratory diseases incidence.³⁴ Whether specifically dedicating public health resources to investigate such cough clusters would increase disease case notifications needs to be investigated. Restricting this cough surveillance analytic approach by monitoring previously diagnosed active pulmonary TB cohorts could identify cough “hotspots” where the risk of TB transmission has been, and may still be, even higher.” (LINE 252--265)

In doing this type of surveillance, it would also serve as a proxy for TB prevalence, as you mention. We therefore mention this at the end of the paragraph: “Temporospatial changes in cough frequency at population-level can be used as a proxy for COVID-19, TB or other respiratory diseases incidence.” (LINE 259-261)

row 178: very preliminary data I would say..

Agreed. We have changed the language to reflect this: “very preliminary data” (LINE 273). We have also added more language that implies that this is at a very early stage and further data is needed: “Further validation is needed using large sample sizes and diverse populations before any definite conclusions can be made regarding their sensitivities and specificities” (LINE 274-276)

row 182-183: could this also be used as a self-screening tool? with subsequent referral to a health facility for TB testing?

Yes. We have included a statement about this at the end of the section: “If deployed on mobile devices, AI-based cough screening could allow for low-cost remote digital active case finding and self-screening, with subsequent referral to a health facility for confirmatory TB testing and linkage to care.” (LINE 282-283) We have also included a short cartoon illustrating how this would occur, using rural India as an example (Figure 3).

row 206: could this be done at the home of a patient by the patient themselves?

It could be done by both. We initially were implying that clinicians can remotely monitor TB patients (providing a patient-centered approach to care), however patients could also be made aware of whether or not they are responding to treatment. We have clarified the how the clinicians would remotely monitor patients and expanded to include the possibility for patients to self-monitor: “cough detection applications could be used as a low-cost, person-centric approach for clinicians to remotely monitor TB patient’s clinical response to treatment, or even for patients to self-monitor their cough as treatment progresses” (LINE 309-312)

row 255: this database that is being developed, it is focused on TB cough or does it have a much wider scope? In general the last section (from row 242 onwards) focusses very much on collecting cough data for TB tool development, which is logical as this is the focus of the perspective. Still, people/patients want to know what the cause is of their cough and when it is not TB they want to know what it is instead. I wonder whether this is something that can be addressed, to take a less vertical view/approach.

The database for now is only funded for TB. However, we agree with the reviewer that we should not promote vertical approaches to care by focusing exclusively on TB AI algorithm development. Other

diseases that cause cough would benefit from such large-scale cough databases. We therefore included a short sentence that acknowledges this in the last section: “Since cough is not specific to TB, such datasets should not be limited to the development of AI algorithms for TB but should also be used to develop and refine cough algorithms for other diseases and conditions that are linked to cough.” (LINE 363-365) We have kept it short since the focus of the perspective is still on TB, as noted by the reviewer.

line 353: reference 21 is incomplete; 27 as well

The references have been updated (note number order in the manuscript has changed):

21: Bales, C. *et al.* Can Machine Learning Be Used to Recognize and Diagnose Coughs? *2020 Int. Conf. e-Health Bioeng. IEEE*, 1–4 (2020).

27: Pahar, M. *et al.* Automatic Cough Classification for Tuberculosis Screening in a Real-World Environment. *Physiol. Meas.* **42**, 105014 (2021).

Reviewer #3 (Remarks to the Author):

A perspective on the potential use of recording of cough analysed by Artificial Intelligence to the detection, specific diagnosis, and monitoring of treatment of tuberculosis is presented. The conclusion is that such application could revolutionise tuberculosis care.

The following comments are for the authors’ consideration

1. The reviewer agrees this is an interesting idea that has been advanced by the COVID-19 epidemic. However, the cited evidence for the use of cough in those circumstances in this article is slim. Perhaps a table of key studies would help more persuasively make the case this technique should be more widely reapplied to tuberculosis.

As mentioned above for reviewer 2, we have included a table in the appendix that summarizes the different cough studies discussed in our manuscript. The table summarizes various topics covered in this manuscript: 1) how objective cough assessment has been used in the context of TB surveillance and care; 2) promising AI tools and algorithms for cough detection; and 3) preliminary study results for COVID-19 and TB cough classification AI algorithms.

2. The authors do not really set out much of a blueprint for research other than to say data collection should be in some way standardized and shared. Yet they mention a few specific applications in detection, diagnosis and treatment monitoring that could be compared with standard of care using rigorous trial designs to determine whether there was incremental advantage. The tuberculosis control world is, in general, not led by early adopters and simply saying something should be massively implemented and analysed because it is possible and cheap may not cut the cheese as evidence for some who determine policy.

The goal of this manuscript is to 1) provide a brief overview of how objective cough monitoring has evolved and 2) describe *hypothetical* applications of cough in the TB care cascade, which we clearly iterate in the text “we outline hypothetical ways in which AI-based cough tools could be used to improve TB outcomes at various stages of the care cascade” (LINE 247-248). It is not meant to be a clear blueprint for researchers, mainly because this field is still in its infancy and there is not enough information available to be making such strong recommendations. Recommendations on

implementation and adoption by the TB community are another topic that need to be addressed subsequently, once we have a clearer understanding of how cough can improve TB care.

3. Line 111 sensitivity for what?

This is the sensitivity for “cough identification”. We have changed “identification” to “detection” to make this more clear (LINE 160).

4. Line 143 – 145 It may not be considered adequate to reference a statement that the TB TimBre has diagnostic utility by mentioning this is based on an unpublished third-party clinical validation. The reader is left wondering what this means and has no means to evaluate this evidence nor even to glean what the perspective authors think of it.

As previously mentioned for another Reviewer comment, the publication for TimBre was recently posted in medrxiv. We have included the reference and updated the information presented to reflect the manuscript. We have decided to include different clinical validation values of the tool TimBre that were published: “In TB, TimBre is a screening application that leverages machine learning to detect TB positive coughs with a sensitivity of 80% and specificity of 92% against a composite reference standard of sputum smear microscopy, GeneXpert (unspecified cartridge), and chest X-ray from a sample of 5 TB-positive and 469 TB-negative individuals” (LINE 214-218)

5. Several spelling mistakes lines (lines 162, 187)

Thank you for pointing these out. We have corrected the typos.

6. Several references incomplete (9, 23, 27, 29, 37, 39)

Thank you for noting this. We have updated the references.

9: Biring, S. S., Prudon, B., Carr, A. J. & Singh, S. J. Development of a symptom specific health status measure for patients with chronic cough: Leicester Cough Questionnaire (LCQ). *Thorax* **58**, 339–343 (2003).

23: Nessiem, M. A., Mohamed, M. M., Coppock, H., Gaskell, A. & Schuller, B. W. Detecting COVID-19 from Breathing and Coughing Sounds using Deep Neural Networks. *IEEE 34th Int. Symp. Comput. Med. Syst.* 183–188 (2021) doi:10.1109/CBMS52027.2021.00069.

27: Pahar, M. *et al.* Automatic Cough Classification for Tuberculosis Screening in a Real-World Environment. *Physiol. Meas.* **42**, 105014 (2021).

29: Schuller, B. W., Coppock, H. & Gaskell, A. Detecting COVID-19 from Breathing and Coughing Sounds using Deep Neural Networks. *arXiv Prepr. arXiv2012.14553* (2020).

37: Davies, G., Boeree, M., Hermann, D. & Hoelscher, M. Accelerating the transition of new tuberculosis drug combinations from Phase II to Phase III trials: New technologies and innovative designs. *PLoS Medicine* vol. 16 e1002851 (2019).

39: Bagad, P. *et al.* Cough Against COVID: Evidence of COVID-19 Signature in Cough Sounds. *arXiv Prepr. arXiv2009.08790* (2020).

Reviewer #4 (Remarks to the Author):

This is an interesting manuscript evaluating the potential role for quantitative cough analysis (using AI)

as a measure to be used in TB clinical care, epidemic control and research. The authors make the argument that cough is crudely utilised at present (usually as a subjective question recorded as a binary variable) and that much valuable information may be lost in the process. They present studies that have been done to assess diagnostic performance of cough for TB screening and diagnosis, but also discuss the limitations of this data and the need for external validation studies. They also call for collaborative efforts to combine data sets and harmonise data collection, to allow AI to be used to further development of this nascent field. They discuss where in the TB cascade cough measures may have an impact. Overall, a useful summary of a research area that needs to be pursued further for its potential.

We thank Reviewer #4 for their detailed summary and enthusiasm.

The authors should discuss the limitation that many patients have subclinical TB and are not coughing – around 50% in a recent prevalence study in South Africa. While they do mention that among people with subclinical TB cough measures may help to detect under-reported cough, its likely that most people in this category do not have a cough. This should be stated as a recognised limitation of cough’s utility of detecting undiagnosed TB in the community.

This is a good point. We have now included a short sentence acknowledging this limitation: “However, digital cough monitoring would not extend to truly asymptomatic individuals with subclinical TB, limiting its application as an active case finding tool in this sub-group.” (LINE 293-295)

For the studies described in line 139-151, it would be useful to provide more details of these studies in a supplementary table – what was the design of these studies, sample size, reference standard, if it was case-control what were controls, etc. It should also be emphasised in this paragraph that early diagnostic studies of novel tests always tend to overestimate diagnostic performance.

As mentioned above for reviewer 2, we have included a table in the appendix that summarizes the different cough studies discussed in our manuscript. The table summarizes various topics covered in this manuscript: 1) how objective cough assessment has been used in the context of TB surveillance and care; 2) promising AI tools and algorithms for cough detection; and 3) preliminary study results for COVID-19 and TB cough classification AI algorithms. We have included information on the study design, sample size, and reference standard.

We also agree with your second comment regarding the tendency for studies to overestimate the diagnostic performance early on. We have acknowledged this with a supporting reference in the following paragraph: “The accuracy of these AI algorithms is contingent on the characteristics of the training dataset. To date, external validation of various AI algorithms has been limited or not been performed and the sample sizes used to evaluate these algorithms have been relatively small. Additionally, early diagnostic studies of novel tests tend to overestimate the diagnostic accuracy, mainly because of the preferential exclusion of more complicated cases. Until sufficient replication studies have been completed using large, and diverse cough datasets representative of different patient populations, the clinical application of these AI algorithms will be restricted.” (LINE 231-238)

Line 171 – its unclear how mapping “cough events from confirmed TB patients” would help with TB control efforts. Please explain this.

Following input from Reviewer #2, we have clarified how syndromic surveillance would be implemented and help with TB control efforts: “Finding people with TB or symptoms of TB requires health systems

and TB programs to strategically deploy limited resources. In a syndromic surveillance approach, both TB susceptible individuals or previously confirmed TB patients could passively and prospectively monitor their cough. Temporal and geospatial aggregations of cough events could in turn be used to better target case finding activities and identify high-risk settings. Temporospatial changes in cough frequency at population-level can be used as a proxy for COVID-19, TB or other respiratory diseases incidence. Whether specifically dedicating public health resources to investigate such cough clusters would increase disease case notifications needs to be investigated. Restricting this cough surveillance analytic approach by monitoring previously diagnosed active pulmonary TB cohorts could identify cough “hotspots” where the risk of TB transmission has been, and may still be, even higher.” (LINE 252-265)

Line 210-211. This assumes that cough is the only mechanism via which the infection is transmitted. There is emerging evidence that other respiratory manoeuvres also transmit TB. This should be noted as a potential limitation.

We agree with the Reviewer and have removed this sentence (LINE 314-317)

There are several spelling errors in the manuscript – eg. line 162

Thank you for pointing these out. We have corrected the typos.

REVIEWERS' COMMENTS:

Reviewer #1 (Remarks to the Author):

The changes made and their justification are fully satisfying and the manuscript appears insightful and informative at this stage.

Reviewer #2 (Remarks to the Author):

I am pleased with the way the authors addressed my comments. The manuscript has improved a lot. I have no further comments.

Reviewer #3 (Remarks to the Author):

The authors have made a reasoned response to mat comments and made some revisions

Reviewer #4 (Remarks to the Author):

Thank you for addressing my comments adequately. I think Table in Appendix is a useful addition